# Intra-annual grounding line migration and retreat: insights from high-resolution satellite and in-situ observations over Milne Glacier in the Canadian High Arctic

5 Yulia K. Antropova<sup>1</sup>, Derek Mueller<sup>1</sup>, Sergey V. Samsonov<sup>2</sup>, Alexander S. Komarov<sup>3</sup>, Jérémie Bonneau<sup>4,5</sup>

Correspondence to: Yulia K. Antropova (YuliaAntropova@cmail.carleton.ca)

Abstract. Changes at the boundary where marine terminating glaciers transition from grounded to floating, known as the grounding line, are critical for glacier stability and ice discharge prediction. We explore changes in the ice flexure zone (FZ), which spans from the inland limit of tidal flexure, known as the hinge line, across the actual grounding line, to where the ice tongue becomes freely floating. FZ movements on Milne Glacier, Nunavut, Canada from 2023 to 2024 were measured at a spatial resolution of 10 m using 4-day RADARSAT Constellation Mission (RCM) repeat acquisitions from four different orbits. The Double Difference Interferometric Synthetic Aperture Radar (DDInSAR) technique was employed to delineate 116 FZ positions, which were then examined with respect to tides, glacier velocity, surface and bed topography. Tides modulated by atmospheric pressure were the major driver of changes in the FZ with the correlation coefficient between DDInSAR and sea surface height displacements ranging from 0.81 to 0.94 for different RCM orbits. The lowest error (<0.1 cm in the SAR line-of-sight) occurred during the 'stable regime', when changes in glacier velocity were minimal, with ascending RCM tracks at incidence angles below 30°. During the 'unstable regime', the glacier velocity became more variable, and the error increased by up to 4 times. In summer, ice in the FZ flowed ~2.7-fold faster than the grounded section upglacier. We discovered a new ~2.4 km grounding line retreat associated with a pinning point at the eastern margin of the glacier. This rapid change is likely due to water intrusion and subglacial melt near the pinning point and shows how these bed features alter the grounding line short-term migration and its long-term retreat.

## 1. Introduction

Marine-terminating glaciers, which drain ice masses from land to the ocean, have experienced dramatic changes associated with a warming climate in recent decades. Acceleration of these glaciers due to rising atmospheric and oceanic temperatures leads to sea level rise and discharge of icebergs or ice islands into adjacent waters (IPCC, 2021). Changes in the areal extent,

<sup>&</sup>lt;sup>1</sup>Department of Geography and Environmental Studies, Carleton University, Ottawa, K1S 5B6, Canada

<sup>&</sup>lt;sup>2</sup>Canadian Centre for Mapping and Earth Observations, Natural Resources Canada, Ottawa, K1A 0E4, Canada

<sup>&</sup>lt;sup>3</sup>Data Assimilation and Satellite Meteorology Research Section, Environment and Climate Change Canada, Ottawa, K1A 0H3, Canada

<sup>&</sup>lt;sup>4</sup>Department of Civil Engineering, The University of British Columbia, Vancouver, V6T 1Z2, Canada

<sup>&</sup>lt;sup>5</sup>Département de Génie Civil et Génie des Eaux, Université Laval, Québec, G1V 0A6, Canada

velocity and calving dynamics associated with Arctic glacier retreat and deterioration were recently documented by Millan et al., (2017; 2023) and Van Wychen et al., (2016; 2020). As a result of these changes, many glaciers located in Greenland and the Canadian Arctic shifted from marine to land terminating glaciers between 2000 and 2020 (Kochtitzky & Copland, 2022). To characterize and explore the stability of marine terminating glaciers, recent studies have focused on the boundary where glaciers transition from grounded to floating (e.g., Rignot et al., 2024; Kim et al., 2024), known as the grounding line. The grounding line exhibits dynamic behaviour, migrating with tides and advancing or retreating in response to changes in glacier dynamics over the long term. Accurate mapping and monitoring of grounding line dynamics are crucial for better understanding glacier retreat patterns and the processes that drive this retreat.

The satellite-based Double-Difference Interferometric Synthetic Aperture Radar (DDInSAR) method has been proven to be efficient for accurate monitoring of grounding line migration with a sequence of SAR images acquired at high temporal (subday to 4 days) and spatial (from sub-metre to a few metres) resolutions (Milillo et al., 2022; Rignot et al., 2024; Antropova et al., 2024; Kim et al., 2024). The DDInSAR method detects the ice flexure zone (FZ) where the glacier surface flexes with tides. This zone, also referred to as the grounding zone in some studies (e.g., Fricker et al., 2009; Freer et al., 2023), spans from the inland limit of tidal flexure (i.e., the hinge line) past the grounding line to the seaward limit of flexure, where hydrostatic equilibrium is reached. However, most recent DDInSAR-based studies consider the hinge line as a proxy for the grounding line, so they tend to refer to the hinge line, and the bounds where it migrates over the tidal cycle, as the grounding line and grounding zone (GZ), respectively (Mohajerani et al., 2021; Chen et al., 2023; Ciraci et al., 2023; Rignot et al., 2024).

To ensure clarity and consistency, hereafter, we use terms 'FZ span' (i.e., the along-glacier distance from the hinge line to the line of hydrostatic equilibrium) and 'GZ span' (i.e., the along-glacier distance from the most landward hinge line to the most seaward hinge line); also referred to as 'grounding zone width' in the recent studies). We also assume that the relative movements (e.g., short-term migration and long-term retreat) of the DDInSAR-derived 'hinge line' and the actual 'grounding line' are equivalent, as their positions are strongly correlated (Rignot, 1996).

Recent studies highlighted the complexity of grounding line migration with tidal cycles and explored causes of this complexity, including glacier geometry, the underlying bed type, subglacial hydrology and sea water intrusions (e.g., Mohajerani et al., 2021; Ciracì et al., 2023, Rignot et al., 2024; Kim et al., 2024). In addition, DDInSAR-based observations showed that GZ span tends to be one to two orders of magnitude larger than predicted by the hydrostatic equilibrium equation (Chen et al., 2023, Milillo et al., 2022, Mohajerani et al., 2021). Since GZs are associated with pronounced ice melt, this implies that the assumption that grounding lines are fixed over space and time made by many conventional numerical glacier/ice sheet models are problematic (Ciracì et al., 2023; Rignot et al., 2024). In order to improve these models and to better understand glacier deterioration processes and long-term glacier retreat, a detailed investigation of the controls that drive grounding line migration over a seasonal cycle is required.

Our study explores intra-annual changes in the Milne Glacier FZ based on a combination of RADARSAT Constellation Mission (RCM) DDInSAR-derived results, in-situ, and model datasets. The objectives of this study are to:

- Delineate the Milne Glacier FZ over short-term migration cycles from February 17, 2023 to September 17, 2024;

70

- Examine the relationship between DDInSAR-derived ice surface vertical displacement and displacement of sea surface height (DSSH). We further relate this correlation to two velocity regimes:
  - a) a stable regime when the in-situ measured Milne Glacier velocity did not change appreciably, and b) an unstable regime when the changes in velocity were marked.
- Explore short-term FZ changes, including its span, its landward position (i.e., hinge line), and the span of the GZ over the study period with respect to DDInSAR-derived surface displacements and glacier surface and bed geometry datasets. In-situ datasets of the Milne Glacier surface and bed geometry, as well as tidal observations, are used to interpret patterns in intra-annual FZ migration. Our results provide information about seasonal migration patterns of the grounding line based on its DDInSAR-derived proxy, the hinge line, and help to elucidate ice deterioration processes that drive grounding line retreat in the long-term.

## 2 Study site and methods

## 2.1 Study site description

Milne Glacier is located on the northern coast of Ellesmere Island, the largest and northernmost island in the Canadian Arctic Archipelago (Fig. 1 inset map). The glacier is one of the major outlet glaciers of the 'Northern Ellesmere Icefield' (Millan et al., 2017) that discharges terrestrial ice into Milne Fiord. In the past, Milne Glacier was the central and most important glacier feeding the Milne Ice Shelf, the second largest remnant of the extensive Ellesmere Ice Shelf that collapsed in the early 20th Century (Mueller et al., 2017). The separation of Milne Glacier from the Milne Ice Shelf was accompanied by the formation and development of an epishelf lake (i.e., a layer of freshwater from summer ice melt and runoff that accumulates above seawater between the shore and an ice shelf) first reported by Jeffries (1984). Evidence of the Milne Ice Shelf thinning associated with the epishelf lake formation was observed from air photos acquired as early as the 1950s (Jeffries, 2002; Mortimer et al., 2012). Field observations of ice thickness and water profiling conducted in 1981 and 1983, respectively, confirmed the presence of an epishelf lake (Prager, 1983; Jeffries, 1985) although this term was not used until Veillette (2008). The area of thin lake ice replacing the thick ice shelf increased from 5 km<sup>2</sup> to 17 km<sup>2</sup> between 1959 and 1984, reaching 28.5 km<sup>2</sup> by 2009 (Mortimer et al., 2012; Mueller et al., 2017). Currently, perennial epishelf lake ice and fragments of the former Milne Glacier Ice Tongue that broke off the glacier between 2009 and 2011 are present within the ~22 km stretch between the present-day floating ice tongue and the ice shelf (Fig. 1 in Antropova et al., 2024). Changes were also observed in Milne Glacier's velocity near the grounding line, which decreased from ~190 m yr<sup>-1</sup> to ~100 m yr<sup>-1</sup> between the 1990s and 2005 (Millan et al., 2017), increased to 120 - 140 m yr<sup>-1</sup> in 2010, slowed down again to 80 - 100 m yr<sup>-1</sup> from 2011 to 2015, and then accelerated to 140 - 160 m yr<sup>-1</sup> between 2016 and 2020 (Van Wychen et al., 2020). The Milne Glacier grounding line retreated by 3.1 km along the glacier's centre line from 1966 to 2023. The average rate of the grounding line retreat was more than twice as high near the western margin (124 m yr<sup>-1</sup>) compared to the centre (53 m yr<sup>-1</sup>) of the glacier between 2011 and 2023.


Warming oceanic masses likely enhanced the grounding line retreat in recent decades since variations in the grounding line retreat rates were associated with changes in oceanic forcing (Antropova et al., 2024). Vertical thinning due to basal melt, was estimated at 15-32 m yr<sup>-1</sup> between 2011 and 2019 (Bonneau et al., 2024) with a greater thinning rate near the glacier's deeper western side than at the shallower central and eastern edges. The glacier's vulnerability to oceanic warming is likely due to its bed geometry (Antropova et al., 2024). Milne Glacier's retrograde bed slope near the grounding line makes the glacier susceptible to marine ice-sheet instability (Schoof, 2007; Weertman, 1974), where grounding line retreat enhances glacier deterioration, which, in turn, triggers further retreat. This positive feedback loop may be underway currently, and will continue until the grounding line reaches a prograde bed slope further upglacier, which will help stabilize the glacier and inhibit further recession of its grounding line.

#### 2.2 Data and methods

Our datasets included satellite SAR and in-situ observations that provide information about the Milne Glacier FZ, the glacier surface and bed topography, ice velocity, and local tides. We used a sequence of RCM single look complex images with HH polarization, acquired from different orbits in 2023–2024, with a nominal resolution of 5 m and a swath width of 30 km. (Table 1). The RCM SAR images were employed 1) to compute changes in the Milne Glacier surface, which allowed delineation of FZ/hinge line and quantification of displacement for the floating ice tongue and 2) to estimate ice velocities.

Table 1. Description of RCM dataset acquired between February of 2023 and September of 2024 for the Milne Glacier FZ delineations and velocity calculations.

| Relative<br>orbit ID<br>and derived<br>results:<br>flexure zone<br>(FZ) and/or<br>velocity (V) | Orbit<br>pass:<br>Des./<br>Asc. | Incidence<br>angle<br>(degrees) | SAR parameters for error calculations (Eq. 2): Incidence angle, Track angle <sup>1</sup> (degrees) | Raw<br>image<br>pixel size<br>Range (m)<br>x azimuth<br>(m) | Period of acquisitions yyyy.mm.dd (time UTC ± 1 min) | Number of<br>FZ<br>delineations |
|------------------------------------------------------------------------------------------------|---------------------------------|---------------------------------|----------------------------------------------------------------------------------------------------|-------------------------------------------------------------|------------------------------------------------------|---------------------------------|
| 9<br>FZ, V                                                                                     | Des.                            | 38.5-40.6<br>(Swath 12)         | 39.5,<br>230                                                                                       | 2.4 × 2.6                                                   | 2023.02.17 –<br>2024.03.23<br>(13:19)                | 42                              |
| 43<br>FZ                                                                                       | Asc.                            | 19.0-21.9<br>(Swath 1)          | 20.3,<br>-58.9                                                                                     | 1.4 × 2.4                                                   | 2023.03.07 –<br>2024.03.17<br>(19:58)                | 27                              |
| 53<br>V                                                                                        | Des.                            | 52.7-54.1<br>(Swath 23)         |                                                                                                    | 2.8 × 2.8                                                   | 2023.03.08 –<br>2024.03.26<br>(12:06)                | -                               |
| 58<br>FZ, V                                                                                    | Asc.                            | 20.1-23.1 (Swath 2)             | 21.5,<br>-58.0                                                                                     | 1.4 × 2.4                                                   | (-=100)                                              | 25                              |





|           |      |                        |                |           | 2023.03.12 –<br>2024.03.26<br>(20:06) |    |
|-----------|------|------------------------|----------------|-----------|---------------------------------------|----|
| 73<br>V   | Asc. | 22.2-25.1<br>(Swath 3) |                | 1.4 × 2.6 | 2023.03.13 –<br>2024.03.27<br>(20:13) | -  |
| 103<br>FZ | Asc. | 26.3-28.9<br>(Swath 5) | 27.5,<br>-55.0 | 1.4 × 2.5 | 2023.03.11 –<br>2024.09.17<br>(20:30) | 22 |

<sup>&</sup>lt;sup>1</sup> Azimuth (clockwise from True North)

To delineate the FZ, the RCM images were processed using the DDInSAR method (Rignot, 1996), with further details described by Antropova et al. (2024). This method reveals the surface displacement of the floating ice tongue along the SAR line-of-sight, by removing the effect of glacier flow, through differencing two consecutive interferograms, under the assumption that the glacier velocity is constant between SAR acquisitions. The hinge line and the hydrostatic equilibrium line are then delineated by examining the landward and seaward boundaries of the fringe pattern associated with glacier flexure. We used the GAMMA software (Werner et al., 2000) to compute the differences in the SAR phase and quantify changes in the surface height between acquisitions. The effect of topography was eliminated using 10 m resolution ArcticDEM (Porter et al., 2023). Interferograms were computed for each pair of RCM acquisitions with a 4-day interval. Then, double difference results were calculated at 10 m resolution for interferogram pairs represented by three consecutive acquisitions (over 8 days) or four acquisitions (over ≥12 days) if there was a gap between acquisitions. The western and eastern FZ boundaries were often affected by noise due to extensive crevassing, which prevented their reliable delineation and the use of the entire area of the FZ for our analysis. Instead, changes to the FZ were quantified as follows.

The FZ boundaries were manually delineated using geographic information system (GIS) software and analysed along three transects on the western, central and eastern sides of the glacier, denoted by subscripts: w, c, E respectively (Fig. 1). Each of these transects have known surface and bed topography from airborne and in-situ radar observations (CReSIS, 2021; Antropova et al. 2024). The transects start up-glacier at an arbitrary fixed point (A) at km 32 along the 2014 IceBridge radar transect. The three transects cross the hinge line at point B and the hydrostatic equilibrium line at point C. We used the following variables to characterize the temporal evolution of the FZ:

- distance to hinge line from the fixed point A (e.g., distances between A and B<sub>W1</sub>, A and B<sub>C1</sub>, and A and B<sub>E1</sub> in Fig. 1);
- GZ span (i.e., the distance along the glacier between the most landward and most seaward hinge line delineations over the period of our observations, e.g., distance between B<sub>W1</sub> and B<sub>W2</sub> in Fig. 1);
- FZ span (e.g., distance between B<sub>W1</sub> and C<sub>W1</sub> in Fig. 1);
  - surface displacement in the vertical direction (i.e., the SAR line-of-sight displacement from the unwrapped interferograms, which was corrected for phase discontinuities along the transects and divided by the cosine of the incidence angle).

Figure 1: Study site. a) Locations of Milne Glacier, in-situ instrumentation, and Global Environmental Multiscale (GEM) grid cell centre. b) Milne Glacier FZ with two examples of its delineation based on SAR images acquired (1) between March 25 and April 10, 2024 (blue polygon) and (2) September 9 and 17, 2024 (orange polygon) shown over a Planet image (Planet Team, 2017) acquired on August 12, 2024. FZ variables explored in this study and computed along the western, central, and eastern transects (subscript W, C, and E) shown in red (through points A, B and C) are: 1) distance to hinge lines (A to Bw1, A to Bc1, A to Bc1 for FZ 1 (blue polygon) and A to Bw2, A to Bc2, A to Bc2 for FZ 2 (orange polygon), 2) span of the two FZs (Bw1 to Cw1, Bc1 to Cc1, Bc1 to Cc1 and Bw2 to Cw2, Bc2 to Cc2, Bc2 to Cc2, 3) GZ span (Bw1 to Bw2, Bc1 to Bc2, Bc1 to Bc2), and 4) surface displacement height of the floating ice.








The vertical component of DDInSAR-derived surface displacement of the floating part of the glacier was analysed with respect to the changes in sea surface height due to tides and atmospheric pressure (Rignot et al., 2024). We used tidal constituents derived from in-situ tidal observations collected between September 3, 2016 and July 4, 2017 (see further details provided in Antropova et al., 2024) to simulate tides for the period of SAR observations using the t-tide toolbox (Pawlowicz et al., 2002). Simulated tide heights were corrected for the inverse barometer effect (IBE) (Wunsch & Stammer, 1997; Padman et al., 2003), using the pressure data recorded by our in-situ weather stations (Fig. 2 c). To calculate the sea surface height (SSH), gaps in the pressure records were filled with the Global Environmental Multiscale (GEM) model data (Buehner et al., 2015). The GEM data were extracted for a GEM grid cell of about 15 km × 15 km centred at 82.6477°N 81.3225°W on the Milne Ice Shelf about 9 km to the northwest of the glacier terminus, so the effect of higher elevations near the grounding line was avoided (Fig. 1). Both, air temperature and surface pressure were in a good agreement with in-situ observations (Fig. 2). DSSH, the difference in SSH for each DDInSAR result, was calculated by subtracting the SSH values at the times of SAR acquisition, consistent with the DDInSAR subtraction. We assessed the relationship between DSSH and DDInSAR-derived surface height displacement based on the correlation coefficient, the root mean squared error (RMSE) and bias. Then, we reconstructed the ice surface height timeseries by least square fitting of our DDInSAR displacements with a single-frequency harmonic function as follows:

$$d(t) = a_1 \sin(\omega t) + a_2 \cos(\omega t) + a_3 H(t) \tag{1}$$

where  $\omega$  is the circular frequency corresponding to the dominant semidiurnal M2 tidal constituent, H(t) is the time-dependant IBE component, and  $a_i$  (i=1,2,3) are the least squares fitting coefficients (Appendix A), t is time of SAR acquisitions. We considered two variants of the fitting function: 1) tide-based only (i.e.,  $a_3=0$ ) and 2) tide-based plus IBE term (i.e.,  $a_3\neq 0$ ). We also explored tidal conditions associated with a new up-glacier notch in the FZ along the eastern transect in detail. We used SSH associated with SAR image acquisition times to calculate the minimum and maximum SSH values, and standard deviation, for each of our DDInSAR results. To characterize the tides associated with notch presence, we calculated average values of these statistics for two groups of DDInSAR grounding line delineations: 1) without the notch and 2) with the new notch present.

To investigate the seasonal effect on the grounding line migration, we split our DDInSAR datasets from all orbits into two periods associated with stable and unstable glacier velocity regimes (Fig. 2 a) for the period of SAR observations. To define these two flow regimes, we used in-situ measured and SAR-derived glacier velocities in the FZ supported by air temperature observations. In-situ velocities were obtained with GNSS receivers (Garbo, 2024) installed near the grounding line (Fig. 1) to record positional information between July 2022 and July 2023. The instruments recorded positional information for three hours daily yielding the GNSS position with an error of about a 1.5 cm after static mode precise point positioning processing (NRCan, 2024). The observed distances between the GNSS positions were converted to daily velocities (Fig. 2 a).



RCM SAR data acquired from four orbits (i.e., relative orbits: 9, 53, 58, and 73; Table 1) were employed to compute ascending and descending range and azimuth speckle offset products in the GAMMA software (Werner et al., 2000). Then, 3D displacement time series were derived from a set of linear equations using inverses (Samsonov et al., 2021). The Milne Glacier velocity represented by the SAR line-of-sight displacement at the GNSS receiver location (Fig. 1) is shown in Fig. 2a. The GNSS-derived velocities agreed well with our SAR-derived results.

The stable and unstable regimes were defined based on the glacier velocity strictly within the period of SAR observations to explore the effect of velocity on our DDInSAR results and don't necessarily represent general patterns in year-round glacier dynamics. Thus, the stable regime was defined as a period of invariant and low glacier velocity of about 0.4 m d<sup>-1</sup> (SAR observations from February to May, 2023 and 2024, a period highlighted in blue in Fig. 2 a, b), which was predominantly associated with winter time and low air temperatures. Conversely, the unstable regime had changing velocity values that decreased from 0.6 m d<sup>-1</sup> to 0.4 m d<sup>-1</sup> (SAR observations from September to December 2023, a period highlighted in orange in Fig. 2 a, b) and followed the summer when positive air temperatures caused ice melt, triggering pronounced glacier acceleration in July and August.

Figure 2: Milne Glacier ice velocity and meteorological observations. a) Daily ice velocity measured with GNSS receivers (black line) and calculated from SAR data (green line) in the FZ. b) Air temperature recorded on Milne Glacier (MG), in Purple Valley (PV), and retrieved from the GEM model. c) Barometric pressure recorded in Purple Valley, overlayed with GEM data, used to correct SSH. The glacier stable regime, highlighted in blue, corresponds to relatively steady and low velocity values of 0.4 m d<sup>-1</sup>. The unstable regime, highlighted in orange, corresponds to velocity values decreasing from 0.6 m d<sup>-1</sup> to 0.4 m d<sup>-1</sup>. DDInSAR observations are shown with triangles: filled if FZ delineation was made, or empty if not, due to low coherence.




The correlation between DSSH and DDInSAR-derived height was calculated for both the stable and unstable regimes and analysed with respect to glacier acceleration computed for the corresponding SAR acquisition times. Acceleration was calculated as a local slope for five GNSS velocity records within a ±2-day time window centred on the day of each SAR acquisition used in the DDInSAR processing. A few missing GNSS records at the end of the unstable regime and beginning of the stable regime in 2024 were linearly interpolated from neighbouring observations.

The error introduced to the line-of-sight surface displacement due to varying ice velocity between SAR acquisitions in the computed DDInSAR displacement, was assessed by calculating the absolute line-of-sight horizontal error,  $\varepsilon_{LOSh}$ ,:

$$\varepsilon_{LOSh} = |\Delta D \times cos\gamma| \tag{2},$$

where  $\Delta D$  is the GNSS-derived glacier difference in glacier displacement between the first and the second SAR image pairs used for the DDInSAR computations,  $\gamma$  is the angle between the 3D-vectors of glacier velocity (taking into account the glacier flow direction of 317.7° with respect to the north direction and the glacier surface tilt of 3.5° in the FZ) and SAR line-of-sight for a corresponding orbit (taking into account the incidence and track angles shown in Table 1) calculated based on Eq. 1 in Samsonov (2024).

FZ variables such as distance from point A (Fig. 1) to the hinge line and the FZ span along the three transects (Fig. 1) were analysed with respect to DDInSAR-derived surface displacement as well as bed and surface slopes retrieved from the airborne IceBridge radar data recorded in 2014 and ground-based ice penetrating radar transects collected between 2016 and 2023 (details described in Antropova et al., 2024). The surface elevation derived from the airborne and ground-based radar datasets was also compared against ArcticDEM (Porter, Claire, et al., 2023), a digital elevation product based on optical stereo imagery collected between 2008 and 2021 over Milne Glacier.

The Milne Glacier GZ span for the entire period of observations (2023 - 2024) was calculated as a difference between the maximum and minimum distances from point A to the hinge line along the three transects from our DDInSAR datasets (e.g., |A B<sub>Wmax</sub>| - |A B<sub>Wmin</sub>| for the western transect). In addition, we assessed the theoretical span of the GZ (ΔL) for the three transects shown in Fig. 1 with the known surface and bed geometry based on the following equation (Tsai & Gudmundsson, 2015):

$$\Delta L = \Delta h_{high} \left( (\alpha - \beta) \frac{\rho_i}{\rho_w} + \beta \right)^{-1} + \Delta h_{low} \left( \frac{\alpha \rho_i}{\rho_w \left( 1 - \frac{\rho_i}{\rho_w} \right)} + \beta \right)^{-1}$$
(3),

Where  $\Delta h_{high}$  and  $\Delta h_{low}$  are the highest (0.50 m) and lowest (-0.44 m) tidal heights corrected for the inverted barometric pressure during the period of observations,  $\alpha$  and  $\beta$  are surface and bed slopes calculated over the entire distance of available in-situ IPR observations and from 38 km to 42 km of the IceBridge transect (Fig. 10),  $\rho_i$  and  $\rho_w$  are ice (917 kg/m<sup>3</sup>) and water densities (1025.4 kg/m<sup>3</sup>, an average value for 100-200 m depth in Milne Fiord), respectively. This equation was derived by Tsai and Gudmundsson (2015) based on prograde bed geometry and was later applied to both retrograde and prograde bed geometries (e.g., Chen et al., 2023). The calculated theoretical span of the GZ was compared against the SAR-derived one. In

addition, radar-derived bed topography in the GZ was used to interpret processes of ice migration and deterioration in this critical zone.

### 3. Results

We delineated 116 ice FZs (Fig. 3 a) using the DDInSAR technique applied to three or four sequential RCM images acquired four days apart from four different orbits. Our delineations revealed short-term fluctuations and systematic changes in the FZs, with the zone's span (i.e., distance along the glacier flow) varying from 46 m to ~2.9 km (Table 2). The results for minimal FZ span were consistent among the four orbits: with the smallest values along the eastern transect, followed by the values for the central transect, and the highest values along the western transect. The maximum FZ span values were consistently higher for the western transect, but values for the eastern transect were often larger than for the central transect. This suggests that ice flexes with tides differently along the three transects, likely due to a distinctive bed geometry and ice thickness.

Table 2. Milne Glacier FZ span over the period of observations.

| Relative orbit ID | Western transect,<br>min – max<br>(m) | Central transect,<br>min – max<br>(m) | Eastern transect,<br>min – max<br>(m) |
|-------------------|---------------------------------------|---------------------------------------|---------------------------------------|
| 9                 | 641 - 2873                            | 485 - 2222                            | 46 - 2422                             |
| 43                | 862 - 2760                            | 548 - 2223                            | 188 - 2153                            |
| 58                | 888 - 2896                            | 620 - 2395                            | 358 - 2522                            |
| 103               | 770 - 2744                            | 561 - 2125                            | 323 - 2574                            |

Our DDInSAR results, based on acquisitions between February and September 2023, re-confirmed that the hinge line was asymmetric with a pronounced notch in the glacier western margin, as reported by Antropova et al. (2024; Fig. 3). However, several interferograms between October 2023 and September 2024 exhibited a new notch in the FZ pointing up-glacier at the glacier's eastern margin. As a consequence, the glacier FZ's eastern edge, that had a much lower average hinge line retreat rate (29 m yr<sup>-1</sup>) than its central (53 m yr<sup>-1</sup>) and western (124 m yr<sup>-1</sup>) regions between 2011 and 2023 (Antropova et al., 2024), experienced rapid changes during the last part of our intensive observation period. Similar to the western side of the glacier, ice deterioration on the eastern side occurred around a pinning point associated with a "bull's-eye" pattern in the DDInSAR interferograms (Fig. 3d).

Figure 3: FZ delineations based on 2023-2024 RCM acquisitions and overlayed on a Planet image acquired on August 12, 2024. a) Orbit 9 delineations: 42 orange polygons. b) Orbit 43 delineations: 27 green polygons. c) Orbit 58: 25 blue polygons. Orbit 103: 22 black polygons overlying a DDInSAR example that reveals two pinning points. A pronounced retreat associated with a new notch on the eastern margin was discovered after hinge line delineations in 2023 by Antropova et al. (2024).

We found that sea surface displacement caused by tides alone (DTide in Fig. 4 a, b, c) had lower correlation with DDInSAR-derived displacement than sea surface displacement caused by both, tides and atmospheric pressure (DSSH in Fig. 4 d, e, f). This is most evident along the western and central transects, where the correlation coefficient (r) range of 0.38 - 0.79 increased to 0.81 - 0.94 when the IBE was included. In general, both RMSE and bias were also lower when atmospheric pressure was

accounted for. The eastern side was not as well correlated since it was mostly grounded throughout the study period yielding a very small DDInSAR-derived surface displacement. These correlation results corroborate the hypothesis that tides, along with atmospheric pressure, govern grounding line migration of Milne Glacier. They also highlight the importance of atmospheric pressure in controlling grounding line dynamics in fjords with relatively small tidal ranges. For Milne Fiord, the simulated tidal range spanned from –20 cm to 21 cm over the observation period. After correcting for IBE, the range increased 2.3-fold, spanning from –44 cm to 50 cm. This change of tidal range resulted in improved correlation with the DDInSAR-derived displacement.

Figure 4: Correlation results calculated for the three transects. Upper panels: correlation between DDInSAR displacement and displacement of simulated tides (DTide) for the: a) western, b) central, and c) eastern transects. Lower panels: correlation between DDInSAR displacement and displacement of sea surface height (DSSH) for the: d) western, e) central, and f) eastern transects. The different orbits are denoted by coloured symbols. 'Cor' - the Pearson correlation coefficient, 'RMSE' - Root mean square error, the 1:1 is plotted as a dashed line.

Unwrapped and corrected surface displacement of the floating ice tongue varied between -40 cm and 60 cm along both western and central transects (Fig. 5 a, b), but this variation was smaller (between -20 cm and 20 cm) along the eastern transect (Fig. 5 c). Thus, most of the time, the eastern part of the glacier ice was grounded with negligible DDInSAR displacement.

Figure 5: An example of DDInSAR-derived surface displacement from orbit 43 along the: a) western, b) central, and c) eastern transects starting at km 32 of the 2014 IceBridge radar profile (i.e., point A in Fig. 1).

The ice surface height time series, reconstructed using the dominant tidal constituent M2 and DDInSAR displacement (Fig. 6 a), highlights the importance of IBE in DSSH calculations. SSH-based reconstruction (Fig. 6 b, c) resulted in a much better agreement with the computed DDInSAR displacement than tide-based reconstruction (Fig. 6 d, e) indicating higher correlation, lower RMSE and bias, respectively.

Figure 6: Reconstructed ice surface height. a) Time-series of ice surface heights, tides, and sea surface heights (SSH, i.e., tides corrected for the inverse barometer effect, IBE). Correlation results for reconstructed displacement and DDInSAR-based displacement calculated for the tide-based reconstruction along the: b) western and c) central transects, and for SSH-based reconstruction along the: d) western and e) central transects. Number of observations, N: 114.

Statistical characteristics of SSH (i.e., sea surface heights at the time of SAR acquisitions rather than their displacement values of DSSH) along the eastern transect for DDInSAR results 1) without the notch, and 1) when the notch was observed, are shown in Table 3. Average minimum SSH conditions were similar (-0.09 m vs. -0.08 m) for both groups of our results, which was also confirmed by the two-sample t-test with high p-value of 0.8. The average maximum SSH was higher for observations associated with the notch presence (0.21 m vs. 0.13 m for observations with and without a notch, respectively). A two-sample t-test resulted in a low p-value of 0.009, confirming the statistical significance of this difference. The standard deviation was

also higher for observations with the notch, than without (0.13 vs. 0.10) since this group was associated with higher maximum SSH values. Thus, with higher SSH, seawater likely intrudes further inland causing a larger FZ and GZ span. Ice displacement and localized flexure around the eastern pinning point creates the a "bull's-eye" pattern in our DDInSAR results.

Table 3. SSH statistical characteristics for the eastern transect.

| Values averaged across all DDInSAR observations | Notch observed | No notch |  |
|-------------------------------------------------|----------------|----------|--|
| Standard deviation (m)                          | 0.13           | 0.10     |  |
| Minimum SSH (m)                                 | -0.09          | -0.08    |  |
| Maximum SSH (m)                                 | 0.21           | 0.13     |  |

The difference between stable and unstable glacier velocity regimes along both, western and central transects for four orbits is presented in Fig. 7. We found that the stable regime was associated with lower mean absolute acceleration (0.02 cm/d² vs. 0.12 cm/d²) than unstable regime, which, in turn, resulted in a lower mean LOSh error (0.4 cm for the stable vs. 1.23 cm for the unstable regimes). In addition, these results were associated with higher correlation between DSSH and DDInSAR displacement (of 0.87 and 0.82) and lower RMSE (11.44 cm vs. 13.22 cm) in the stable rather than in the unstable regime, respectively.

Figure 7: DDInSAR-derived displacements versus DSSH combined along the western and central transects with respect to the two glacier velocity regimes. a) Stable regime with minimal changes in velocities and b) unstable regime with more pronounced changes in velocities as defined in Fig. 2.

When these results were analysed by orbit (Fig. 8), we again found that mean absolute acceleration and mean LOSh error values were consistently lower for the stable regime than for the unstable one for all four orbits. The ascending orbits with low incidence angles (i.e., orbits 43, 58, and 103) showed high correlation between DSSH and DDInSAR displacement (0.91 – 0.95 in Fig. 7 c, e, g) for the stable regime, which was higher than correlation shown for the unstable regime (0.54 – 0.8 in Fig. 8 d, f). RCM orbit 43 results were associated with 11 times higher deceleration during the unstable than stable regime (-0.11 cm/d<sup>2</sup> vs. 0.01 cm/d<sup>2</sup>), which resulted in the lowest correlation (0.54) between DSSH and DDInSAR displacement. Descending

orbit 09, with a high incidence angle, exhibited the opposite: a higher correlation during the unstable than during the stable (0.93 vs. 0.83 in Fig.8 a, b) regime.

Figure 8: DDInSAR-derived displacements versus DSSH along the western and central transects with respect to the glacier velocity regimes as defined in Fig. 2. Stable regime with minimal changes in velocities for four obits and low LOSh errors: a), c), e), g); and unstable regime with more pronounced changes in velocities and high LOSh errors for four orbits: b), d), f), h).

Our analysis indicates that changing glacier velocities contributed a minor LOSh error to our DDInSAR results (

Figure 9: Association between DDInSAR surface displacement and FZ variables along the western, central, and eastern transect. a) Distance from Point A to hinge line (HL) and DDInSAR-derived surface displacement overlayed by lines of linear best fit. b) FZ span along the three transects with respect to DDInSAR surface displacement overlayed by lines of linear best fit.

The DDInSAR-derived GZs shown with vertical red dashed lines in Fig. 10 b-d were all associated with retrograde bed slopes. The GZ span was 2.3 km along the eastern, 1.9 km along the central and 2.6 km along the western transects (Fig. 10), which was about 55, 22 and 74 times larger, respectively, than calculated from the hydrostatic equation (Eq. 3). The western transect (Fig. 10 d), associated with the shallowest bed slope, had the largest GZ span, and the central transect (Fig. 10 c) associated with the steepest slope had the smallest GZ span, which is aligned with the theory (Tsai & Gudmundsson, 2015) and literature (e.g., Rignot et al., 2024). In addition, our DDInSAR results suggest the presence of a pinning point farther westward of the bed rise ('BR3' in Fig. 10 d) between km 38 and km 39 of the western transect. This likely further amplifies the effect of the shallow slope by bridging the ice to the area where the bathymetry becomes shallower upglacier of the tributary glacier that merges with Milne Glacier. The eastern bed slope was also associated with the pinning point shown by our DDInSAR results, which extended between the two bed rises at km 39.7 and km 40.7 highlighted by grey colour in Fig. 10 b. This pinning point altered the bed slope and the span of the observed GZ. Additionally, the eastern pinning point was linked to a pronounced retreat of the grounding line over the last portion our SAR observation period, caused by seawater entering a cavity upglacier of the pinning point. The eastern transect bathymetry suggests that seawater intrusion further upglacier is likely currently inhibited by a small bed rise ('BR1' in Fig. 10 b) at km ~38.7 and could be later suppressed by a larger bed rise at km ~38.2 ('BR2' in Fig. 10 b), that will likely become the next landward position for the grounding line. Subsequently, another cavity on a retrograde slope may cause a rapid retreat of the grounding line down to km 37.5, where the bed slope changes to a prograde slope, which should slow down the grounding line's recession.

Figure 10: Retrograde bed and surface geometry of the Milne Glacier near its GZ. a) The most landward FZ boundaries delineated in September, 2024 (blue polygon) overlaying the three transects and a Planet image from August 12, 2024. Surface and bed geometry retrieved from the field radar observations and Arctic DEM along and in vicinity of b) eastern, and c) central transects. The calculated span of the GZ (displayed in text) is about 22 to 74 times smaller than the DDInSAR-derived GZ (shown with red dashed lines).

Over the period of our SAR observations, the Milne FZ was associated with velocities of up to 270 m yr<sup>-1</sup> during the summer (Fig. 11). The grounded ice moved downstream at about 100 m yr<sup>-1</sup> and then accelerated to 200 m yr<sup>-1</sup> at ~3 km up-glacier of the FZ, reaching a velocity of 270 m yr<sup>-1</sup> in the FZ and then up to 320 m yr<sup>-1</sup> down-glacier of it during summer months. Hence, Milne Glacier FZ zone was associated with ~2.7-fold increase of glacier velocity compared against its upglacier grounded part shown in Fig. 11.

Figure 11: SAR-derived Milne Glacier velocity along the a) eastern, b) central, and c) western transects with respect to its GZ and the largest span of FZ marked by white and yellow dashed lines respectively over period of observations. Profiles start at km 32 of the 2014 IceBridge radar transect. (point A in Fig. 1). Velocity fluctuations between km 32 and 37 in the summer are a noise-related artifact.

The landward part of the FZ, associated with the GZ where the GZ migrated during the observation period, exhibited a pronounced shift in velocity, as this is the zone where ice flow becomes frictionless. The largest GZ span along the western transect of the glacier (Fig. 11 a) was associated with higher velocity, discharging more ice than its central and eastern transects (Fig. 11 b, c).

#### 4. Discussion





This section summarizes our results and insights about the controls that drive short-term grounding line migration taking into account variations in glacier velocity and different SAR viewing geometry. Our observations also reveal patterns that contribute to long-term grounding line retreat.

# 4.1 Short-term migration of GZ and FZ: tides, atmospheric pressure, and glacier geometry

The intra-annual DDInSAR-derived vertical displacement of the Milne Glacier ice tongue is governed by changes in the SSH, which in turn are driven by variations in tides and atmospheric pressure. While sea surface height (SSH) has been identified as a major driving mechanisms of grounding line migration for Thwaites Glacier in West Antarctica (Rignot et al., 2024), Fisher, Mellor, and Lambert glaciers in East Antarctica (Chen et al., 2023), and Jakobshavn Isbræ in Greenland (Kim et al., 2024), our study highlights the important role atmospheric pressure can have in this process. Changes in atmospheric pressure result in an isostatic response of sea surface height (i.e., the inverse barometer effect), which then affects ice-shelf surface elevation (Padman et al., 2003). For marine-terminating glaciers systems similar to Milne Glacier, where tidal range is much smaller than in the aforementioned studies, atmospheric pressure has a pronounced effect on SSH, and consequently, on the displacement of the floating ice tongue observed from the DDInSAR results.

Our findings corroborate the idea that shallow bed topography is associated with a larger span of GZ (Milillo et al., 2022; Rignot, 2024) than steep bed. The grounding line of Milne Glacier migrates along a retrograde bed slope over tidal cycles forming distinctive GZs along the three transects with known geometry. The largest GZ span was observed along the shallowest western transect followed by span over eastern transect, and then over central transect with the steepest bed slope.

The overall bed slope in the GZ helps to characterize its span, but local topography also plays an important role. For instance, Chen et al. (2023) used intervals associated with very distinctive bed features for their slope calculations (i.e., an interval within the Lambert Estuary's GZ associated with retrograde slope and several intervals upglacier of Fisher Glacier's GZ associated with very steep prograde slopes) to demonstrate a quasi-exponential relationship between GZ span and bed slope. Milne Glacier's topography within its GZ associated with pinning points drastically affected GZ span along the eastern transect over the period of observations: the grounding line migrated about 800 m upglacier (35% of the observed GZ span) when seawater intruded into a cavity upglacier of the pinning point. The grounding line migration was even more extensive farther east, with landward incursion reaching up to 2.4 km further than the previous GZ. Thus, pinning points on retrograde slope play an important role in grounding line migration and shaping GZ characteristics.

The relationships between DDInSAR vertical displacement and FZ variables such as distance to hinge line and FZ span were examined to explore patterns in the ice FZ with changes in SSH. Our results show that moderate linear relationships exist between DDInSAR displacement and distance to hinge line as well as FZ span. However, pinning points likely influence these relationships by serving as anchors for the floating ice tongue. They inhibit seawater intrusion into bed cavities during low




SSH and permit water intrusions farther inland when SSH is high. Therefore, detailed information on bed geometry in the GZ and farther upglacier is essential to enhance our understanding and improve predictions of grounding line retreat.

## 4.2 Migration during stable and unstable velocity regimes

Our analysis provides insights about the robustness of DDInSAR-derived results with respect to the RCM SAR viewing geometry and two velocity regimes, stable and unstable, defined based on in-situ velocity observations. The calculated mean LOSh errors were below 1 cm for both regimes and all orbits except for orbit 09 during the unstable regime, which yielded a mean LOSh of 2.27 cm. We found that the RCM viewing geometry associated with low incidence angles (between 19° and 30°) and ascending tracks (i.e., orbits 43, 58 and 103) provided the most robust results for the Milne Glacier FZ delineation during the stable regime. These results degraded with increasing changes in velocity, especially for orbit 43, when the glacier decelerated by a factor of 11 (-0.01 cm/d² vs. -0.11 cm/d²) LOSh error increased 4-fold (0.06 cm vs. 0.24 cm).

The descending RCM SAR pass (i.e., orbit 09) with a mean incidence angle of 39.6° had the highest mean LOSh error. However, the correlation between DDInSAR and DSSH was high during both, stable and unstable regimes (0.83 and 0.93, respectively). We attribute this result to the different viewing geometry (i.e., descending path and high incidence angle), which likely affected the correlation, as well as to the least well-defined distinction between the stable and unstable seasons among the considered orbits. The glacier decelerated only by a factor of ~4 (-0.03 cm/d² vs. -0.13 cm/d²) and LOSh error increased 2-fold (0.95 cm vs. 2.27 cm). Our investigation was limited by the number of FZ delineations during the unstable season due to relatively scarce summer SAR observations as well as lack of coherence during the summer season caused by both surface melt and acceleration of the glacier. With the increasing number of SAR observations, larger SAR datasets along with reliable in-situ tidal and velocity observations, could help to further investigate and quantify how the grounding line and FZ delineation is affected by the SAR viewing geometry with respect to changes in glacier velocity.

# 4.3 Glacier FZ velocity

The glacier FZ plays a crucial role in ice dynamics, as it controls the discharge of ice from the grounded portion of the glacier (Schoof, 2007). The Milne Glacier's FZ and the rest of its floating ice tongue exhibited higher velocities than the grounded portion of the glacier, with velocities increasing by up to 2.7-fold as the glacier transitioned from grounded to floating in the summer. Moreover, the GZ with the largest span along the western transect showed higher velocities throughout our period of observations suggesting that a larger GZ span associated with a shallow bed geometry might promote increased ice flow into the ocean. The observed expansion of the GZ along the eastern transect might indicate future velocity increase in this part of the glacier. The computed average annual velocity varied from 170 to 195 m yr<sup>-1</sup> in the FZ during the study period was similar to results for 1990s reported by Millan et al., (2017) and higher than observed velocities over recent years. Between 2010 and 2020, the glacier velocity in the FZ ranged from 80 to 160 m yr<sup>-1</sup> with its maximum happening between 2018 and 2020 (Wychen et al., 2020). This study documented glacier acceleration, which was preceded by grounding line retreat, suggesting






that the glacier is prone to dynamic thinning that could trigger further grounding line retreat. The newly observed notch in the eastern part of the glacier might suggest future changes in the glacier dynamics.

### 4.4 Long term retreat and bed geometry

The Milne Glacier retrograde bed configuration in the GZ allows ocean water to intrude landward with high tides behind two pinning points in its western and eastern margins and, likely enhances ice deterioration in this zone which would promote grounding line retreat. For instance, Poinelli et al. (2023) suggested that enhanced melt rates around pinning points can lead to the mechanical instability of an ice shelf/tongue. The observed rapid grounding line retreat in the eastern part of Milne Glacier illustrates the combined effect of bathymetry and ocean forcing. Between 2011 and October of 2023, this part of Milne Glacier was stable and had much smaller span of the GZ than western and central parts (Antropova et al., 2024). The eastern GZ was also associated with a bed rise on a retrograde slope (before km 41 in Fig. 10 b), which, likely, inhibited the seawater intrusion further inland. As a result of melt processes during summer 2023, the grounding line migrated past the next inland bed rise shown as the landward bound of pinning point region in Fig. 10 b, and the new eastern landward limit of the FZ was observed up to 2.4 km upglacier at 'BR1' in Fig. 10 b. This suggests rapid changes as soon as seawater was able to intrude past the first bed rise, likely driven by the retrograde slope and permanently opened ocean cavities.

While tides allow for the renewal of the water inside the cavities around the pinning points, the glacier uplift caused by tides and atmospheric pressure also allows periodic seawater intrusions everywhere in the GZ. On Petermann Glacier, 400 km southeast of Milne Fiord, Ciraci et al. (2023) estimated submarine melt rates up to 80 m/yr in the GZ due to seawater intrusions. Sea water intrusion is also thought to be an important component of the recent retreat of Thwaites (Rignot et al. 2024) and Jakobshavn Isbræ (Kim at al. 2024). At Milne Fiord, where SSH range is about 0.41 m, a 2.4 km and a 1 km incursion along and across the eastern part of the fjord, respectively, corresponds to approximately 11.4 m³/s of seawater moving in (during flood tide) or out (during ebb tide) of the Milne Glacier GZ. Given that this water is roughly 1°C above the local freezing point at 150 m depth (Hamilton 2021; Bonneau et al., 2024b), these intrusions have the potential to melt about 0.15 m³/s of ice. This is comparable to the estimated submarine melt rate along the entire ~5 km-wide submerged glacier front, which is 0.82 m³/s (Bonneau et al., 2024a), underscoring the potential effect of seawater intrusions in Milne Glacier GZ retreat.

Our DDInSAR observations of Milne Glacier reveals that its grounding line retreat along the western and eastern transects were driven by the sea water intrusion upstream of the pinning point, which can lead to a 10%–50% increase in the volume loss from marine-terminating glaciers and ice sheets (Robel et al., 2022). The dramatic Milne Glacier grounding line 1992-2023 retreat along its western margin started with a notch, which was associated with a pinning point and seawater intrusion behind it (2011 DDInSAR results in Antropova et al., 2024). This notch widened towards the glacier centre between 2011 and 2023. The present study shows that the eastern margin has recently also been experiencing an extensive ungrounding around a pinning point, with a new notch likely expanding towards the glacier centre. Thus, the SAR-derived patterns of the grounding line retreat suggest that ice ungrounding progresses from pinning points towards the glacier centre driving ice deterioration

and further grounding line retreat. Comprehensive in-situ observations of ice thinning and melt processes near pinning points combined with detailed bed information and frequent satellite DDInSAR observations would provide additional details and insights about processes causing ice deterioration near the pinning points.

### 485 Conclusions




Our study demonstrated the benefits of the RCM 4-day high-resolution coherent change detection acquisition cycle for monitoring of glacier FZs and grounding line migration at a high temporal resolution from four different orbits. These relatively frequent RCM acquisitions spanning from February 2023 to September 2024 allowed us to examine a large SAR dataset to better understand the Milne Glacier FZ intra-annual migration pattern, while also taking into account the SAR viewing geometry. In addition, we were able to support our DDInSAR observations with in-situ glacier-specific and environmental datasets and explore the presence of a seasonal component related to glacier stable and unstable velocities in our observations. Our results showed that Milne Glacier FZ intra-annual migration is driven by tidal cycles and atmospheric pressure, the latter of which can play an important role for fiords with small tidal ranges. DDInSAR results were reliable despite velocity variations during the unstable season, but the effect of viewing geometry requires further investigations. Bed geometry, specifically pinning points, play a crucial role in shaping the FZ and governing grounding line retreat. In-situ observations of melt processes near pinning points could provide further insight into their influence on grounding line dynamics.

Continuous frequent satellite-based monitoring of changes in the FZ and GZ of marine-terminating glaciers, such as our work here with RCM SAR, serves as an efficient tool to explore short-term migration of these crucial zones. Consistent and frequent SAR observations over Arctic and Antarctic glaciers in combination with improved datasets on their bed geometry and ice thinning will ultimately improve our understanding of ice deterioration in the GZ that, in turn, will improve our glacier modeling capabilities.

# Appendix A

We assume that the glacier vertical displacement can be described by the single-frequency harmonic function given by Eq. 1. Then, each observed SAR-derived double difference displacement,  $\Delta d_i$  (i is the number of observations: 1, 2, ... n) is given by the following linear combination with unknown coefficients  $a_k$  (k = 1,2,3):

$$\Delta d_i = a_1 S_i + a_2 C_i + a_3 H_i, \tag{A1}$$

where


$$S_i = [\sin(\omega t_2^{(i)}) - \sin(\omega t_1^{(i)})] - [\sin(\omega t_4^{(i)}) - \sin(\omega t_3^{(i)})], \tag{A2}$$

$$C_{i} = [\cos(\omega t_{2}^{(i)}) - \cos(\omega t_{1}^{(i)})] - [\cos(\omega t_{4}^{(i)}) - \cos(\omega t_{3}^{(i)})], \tag{A3}$$

$$H_{i} = [H(t_{2}^{(i)}) - H(t_{1}^{(i)})] - [H(t_{4}^{(i)}) - H(t_{3}^{(i)})],$$
(A4)

where  $t_j^{(l)}$  are known times of RCM SAR acquisitions represented as serial date numbers (e.g., the MATLAB notation of 'datenum') and  $\omega$  is the circular frequency corresponding to the dominant semidiurnal M2 tidal constituent.

For the set of *n* DDInSAR observations, the following system of linear equations can be obtained:

$$MA = D, (A5)$$

where:

$$M = \begin{bmatrix} S_1 & C_1 & H_1 \\ S_2 & C_2 & H_2 \\ \vdots & \vdots & \ddots & \vdots \\ S_n & C_n & H_n \end{bmatrix}, D = \begin{bmatrix} \Delta d_1 \\ \Delta d_2 \\ \vdots \\ \Delta d_n \end{bmatrix}, A = \begin{bmatrix} a_1 \\ a_2 \\ a_3 \end{bmatrix}.$$
 (A6)

Then, the unknown coefficients are derived using the least squares method as follows:

$$A = (M^T M)^{-1} M^T D. (A7)$$


**Data availability:** The authors do not have permission to share raw RCM data, the IceBridge airborne radar data are freely available, other field and modeled data can be shared by request.

**Author contribution:** YA: collected field data, developed methodology and algorithms to conduct the data processing and analysis, participated in the funding acquisition, prepared the initial manuscript; DM: supervised the research project, acquired funding, managed field work, collected data; SS: developed software and conducted SAR data processing; AK: tasked RCM satellites, participated in the data analysis; JB: collected and processed field data. All: conceptualized the research project, reviewed, and edited the manuscript.

**Competing interests**: The authors declare that they have no conflict of interest.

Acknowledgements: We are thankful to the Polar Continental Shelf Program for the logistical support that greatly facilitated all field work activities related to this study, and to the Research Computing Services at Carleton University for IT support. We thank all members of the 2014, 2016, 2019, 2022, and 2023 field campaigns in Milne Fiord. We are grateful to Adam Garbo for building the GNSS instruments. We thank the Canadian Space Agency for providing the RADARSAT Constellation Mission (RCM) imagery. The airborne radar data used in this study were acquired by NASA's Operation IceBridge. We acknowledge the use of data products from CReSIS generated with support from the University of Kansas, NSF grant ANT-0424589, and NASA Operation IceBridge grant NNX16AH54G.

**Financial support:** This project was supported by the: Natural Sciences and Engineering Research Council of Canada (YA: CGS M 2019, and PGSD3–546559-2020; DM: Discovery Grant and Northern Research Supplement RGPIN-2016-06244, RGPNS-2022-04875), W. Garfield Weston Foundation (YA: 2019–2020 Northern Research and Extended Stay Awards), Canada Foundation for Innovation (DM:31410), Ontario Research Foundation (DM: 31410), Polar Continental Shelf Program (DM: 627–18; 651-14, 636-16, 651–19, 665-22, 655–23), ArcticNet (a Network of Centres of Excellence of Canada) GO-Ice project (YA,DM, and JB), Polar Knowledge Canada (YA and JB: Northern Scientific Training Program 2019, 2022, and 2023).

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
