# Peer review of "Intra-annual grounding line migration and retreat: insights from high-resolution satellite and in-situ observations over Milne Glacier in the Canadian High Arctic"

_EGUsphere, 2025_

## Referee Comment (RC2)

Review of Antropova et al., submitted to The Cryosphere Discussion

**Intra-annual grounding line migration and retreat: insights from high-resolution satellite and in-situ observations over Milne Glacier in the Canadian High Arctic**

**By Christian T. Wild**
January 2026

**Summary:**
Antropova et al. present a detailed differential interferometric SAR (DInSAR) analysis of the tidal-flexure zone of Milne Glacier. Using a high-temporal-resolution RADARSAT dataset with 4-day repeat intervals, combined with in situ observations, they test the commonly assumed steady-flow condition applied in many DInSAR grounding-line studies. They also show that intra-annual grounding-line retreat occurs in a spatially and temporally heterogeneous pattern. They link this variability to subglacial topography and the development of localized pinning points, and offer speculations about the implications for basal melt rates and the potential for upstream ocean-water intrusion beneath the ice.

Strengths:
The authors demonstrate strong command of the DInSAR technique from a remote-sensing perspective and reveal clear, instructive examples of well-known processes associated with tidal flexure of ice shelves, which I elaborate on below. I have rarely seen such a thorough quantification of the steady-flow assumption, nor such careful consideration of how different satellite tracks (and associated variations in incidence angle and azimuth) capture the tidal-flexure signal. For this, I commend the authors. I particularly appreciate their use of available GNSS data for validation. However, because the GNSS measurements may be temporally biased (limited to a three-hour acquisition window per day, L183, which may also explain why they yield substantially higher ice velocities than the SAR-derived estimates in Fig. 2a), I wonder whether the analysis should be extended to include SAR-derived velocities alone. This would make the approach more transferable to settings where GNSS data are unavailable and could provide a clearer spatial picture of how the steady-flow assumption performs across the tidal-flexure zone, where velocity gradients are most pronounced towards their lateral margins.

Weaknesses:
The study lacks a clearly defined glaciological hypothesis and instead reads largely as an enumeration of observations that have been explored extensively in the past.. Similar datasets have already been used to address, among other things: (1) improvements to tidal models (Wild et al., 2019), (2) the depth of the neutral layer in a deforming beam, (3) effective ice stiffness (Wild et al., 2018), (4) shear-margin strength (Wild et al., 2025), (5) two-dimensional effects on flexural patterns (Rack et al., 2017; Wild et al., 2018). Many of these applications have been published in this journal, yet the authors appear to be insufficiently engaged with this literature. This is reflected in an incomplete reference list, shortcomings in the introduction, and at times imprecise or incorrect terminology.

Given these issues, and considering that most recent glaciological DInSAR studies in *The Cryosphere* now combine DInSAR with multi-sensor approaches (Li et al., 2023), machine-learning methods (Ramanath et al., 2025), broad spatial (Picton et al., 2023) or temporal coverage (Chien et al., 2026), or numerical modeling of tidal flexure (Ross et al., 2024; Wild et al., 2025), I recommend that this manuscript may be more suitably placed in a more method-focused journal such as IEEE or a similar outlet. In that context, it would serve as an excellent technical reference that showcases the paper's considerable methodological strengths. If the manuscript were to be framed more explicitly within a glaciological scope, it could be strengthened by a robust investigation of short-term ice-flow dynamics and their relationship to changing buttressing as pinning points emerge and evolve.

**Major comments:**

1) The bending effect introducing spurious signals in the tidal-flexure zone

[Figure]

Figure R1: (left) from Wild et al., 2018, (right) from Rack et al., 2017. Red illustrates compression, blue horizontal extension. Arrows indicate magnitude and direction of geometric displacement around a neutral-layer.

The interpretation of individual DInSAR observations warrants more careful consideration of the vertical structure of tidal flexure. These measurements are inherently limited to the ice surface, whereas flexural deformation occurs about a neutral layer located approximately midway through the ice thickness (as illustrated in Fig. R1). In that schematic, the upper part of the beam moves to the left (red) while the lower part moves to the right (blue), highlighting that the ice deforms in opposite directions above and below the neutral layer. Vice versa for low tide as also illustrated. As a result, the surface experiences a harmonic back-and-forth motion during

the tidal cycle, which can generate an apparent, spurious grounding-line migration on the order of one local ice thickness (Rack et al., 2017), which is within the y-range of the scatter presented in Fig. 9a.

This effect is further complicated in two dimensions by grounding-line sinuosity (Wild et al., 2018), which modifies the surface flexure pattern relative to a simple one-dimensional beam (it's dampened within embayments, and propelled around protrusions of the grounding line). In principle, this "bending effect" can be quantified and subtracted using a simple elastic model (with the temporal mean grounding line position, an ice thickness map and the DSSH). The tidal-flexure slopes in both the x and y directions are then combined with assumptions about the distance to the neutral layer. The corresponding x and y components of the bending effect can then be rotated into the radar line-of-sight using straightforward trigonometric relationships.

$$\int_x \delta_x \, dx = Be_x = \frac{H}{2} \, w'_x$$
$$\int_y \delta_y \, dy = Be_y = \frac{H}{2} \, w'_y.$$

Given the importance of this effect for grounding-line interpretation, the manuscript would benefit from a more explicit treatment of how surface flexure, the position of the neutral layer, and two-dimensional grounding-line geometry influence the inferred grounding-line signal.

In particular, I am concerned that the delineation of the tidal-flexure zone may be partly biased. Support for this interpretation is provided by the observation that DInSAR-derived displacements acquired at higher incidence angles show improved correlation during the unstable regime (Fig. 8a/b and L319–321). As the radar incidence angle increases, the sensor becomes more sensitive to horizontal surface motion and its temporal variability, which amplifies the contribution of geometric surface deformation associated with flexure, compared to a lower incidence angle.

This also has implications for the authors' conclusion that "a larger grounding-zone width promotes increased ice flow" (L443-445). Thicker ice is expected to exhibit a more pronounced bending effect than thinner ice, and the associated geometric surface motion should be greatest at the grounding line itself, where curvature is most pronounced. This could, at least in part, create or exaggerate the appearance of a wider grounding zone independent of dynamic ice flow.

For this reason, the multi-incidence-angle DInSAR dataset presented in the manuscript offers a valuable opportunity. The differing sensitivity of the observations to horizontal versus vertical motion provides a promising, and untapped, means of constraining the depth of the neutral layer beneath the ice surface. Such an analysis would also represent a significant contribution to ongoing glaciological efforts to infer basal melt rates from phase-sensitive radar measurements in the tidal-flexure zone (Vaňková et al., 2020).

2) Unraveling (here called reconstruction) of DInSAR to single tides

The authors correctly note that DInSAR images represent a double difference of three or four consecutive SAR acquisitions, each taken at different phases of the tidal cycle and therefore containing contributions from multiple tidal constituents as well as atmospheric effects. Consequently, a direct comparison between tide models and DInSAR observations requires either that the same SAR differencing procedure be applied to the modeled tides (as in Fig. 4), or that the DInSAR signal be decomposed into individual tidal components (as attempted in Fig. 6).

The authors pursue the latter approach by fitting a single harmonic with a prescribed angular frequency corresponding to one dominant tidal constituent, both with and without the inclusion of the inverse barometer effect (IBE). While the fit does improve when the IBE is included (Fig. 6), this improvement is expected and has been demonstrated in numerous previous work (e.g., Rignot et al., 2000; Padman et al., 2003; Wild et al., 2022). Importantly, the remaining root-mean-square error (~13 cm; Fig. 6b/c) remains large relative to the total displacement range of approximately −40 to +60 cm in the corresponding double-difference interferograms (Fig. 5a). In my view, this relative magnitude of residual error limits the physical interpretability of the harmonic decomposition using a single sinusoid. I would also like to draw the author's attention to (Minchew et al., 2017), who underscore the importance of using all sinusoids for the unraveling/reconstruction (with former knowledge of their phase from GNSS).

In previous work, we developed and applied an alternative approach for unraveling DInSAR to single tides  for Darwin Glacier (Wild et al., 2019) and subsequently for Priestley Glacier (Wild et al., 2025), both published in *The Cryosphere*. This method uses the mean tide-deflection ratio (the so-called alpha map first introduced by Han and Lee, 2015) to separate DInSAR observations into individual tidal components, reducing the residual RMSE to a level comparable to interferometric noise (on the order of 1 cm for Sentinel-1). Based on the flexure curves shown in Fig. 5, the Milne Glacier dataset appears well suited for this type of analysis, and its application here could substantially strengthen the study and its glaciological scope. In this context, relying solely on a single-harmonic fit, despite recognizing the importance of IBE forcing, seems a missed opportunity to fully exploit the dataset. I would be very willing to discuss or advise on such an analysis if helpful, while recognizing that this would extend beyond the minimum expectations for a reviewer.

Two additional benefits of the alpha-map approach are worth noting. First, it naturally averages over the bending effect associated with the position of the neutral layer, because DInSAR observations sample all phases of the tidal cycle (as illustrated in Fig. 5). Second, it mitigates the influence of viscoelastic effects, which appear to be present in several flexure curves along the West and Central transects, through temporal averaging (discussed further below), so it avoids the implementation of a numerical model.

3) Tidal forcing and flexure-zone width

I would appreciate further clarification regarding the physical basis for the reported relationship between flexure-zone width and the magnitude of the tidal forcing. From my understanding, for

a given tidal forcing, the width of the flexure zone should primarily depend on ice stiffness, which is itself governed by ice thickness and the effective Young's modulus (Fig. R2). Under this framework, the magnitude of the tidal forcing should affect displacement amplitudes, but not the spatial extent of the flexural response. Similar for ice viscosity, which affects the timing and shape of the flexure, but not its horizontal extent.

[Figure]

Fig. R2: from Wild et al., 2017. Both panels show that for a given tidal forcing (here -0.6 m) and ice thickness, the flexure-zone width is only dependent on the Young's Modulus. Ice viscosity affects the shape and timing of the flexural wave.

For this reason, I am concerned about the interpretation presented in Fig. 9b, as well as the conclusions drawn from it, may not be physically well founded. In addition, it is unclear whether the linear relationships shown are statistically significant. While some structure is visible in the scatterplot, this pattern may alternatively reflect temporal aliasing in the SAR-data acquisitions, potentially related to tidal phase or inverse barometer effects at the time of SAR-data collection.

To explore this possibility, it would be helpful if the authors could reproduce the classification used in Fig. 2a (i.e., the different triangle symbols) in Fig. 6a and explicitly assess whether systematic biases arise from the timing of the SAR-data acquisitions relative to tidal forcing and IBE. Such an analysis would help to clarify whether the observed relationship reflects a physical signal or an acquisition-related artifact.

4) Tidal forcing and grounding-zone width

Furthermore, the manuscript presents a similar analysis relating tidal forcing to grounding-zone width (referred to as the "Distance to HL" in Fig. 9a). This type of correlation implicitly assumes that tidal flexure responds instantaneously to the forcing, i.e., that the ice behaves purely elastically. However, at tidal frequencies, glacier ice is well known to exhibit viscoelastic behavior (e.g., Reeh et al., 2003; Wild et al., 2017). Under such conditions, a time lag between the tidal forcing and the flexural response is expected. Importantly, this delay is not constant but is anticipated to increase nonlinearly from the freely floating portion of the ice shelf toward the grounding (hinge) line.

This viscoelastic response provides a plausible alternative explanation for the observed negative correlations. Time delays on the order of tens of minutes to several hours have been reported in similar settings and should be carefully accounted for when interpreting correlations between tidal forcing and grounding-zone metrics. In this context, applying a simple linear fit across all observations may not be physically appropriate.

A useful first step toward quantifying this time delay would be to correlate the available GNSS observations acquired within the tidal-flexure zone with an independent tide model representative of the freely floating ice shelf. For example, the Gr1kmTM model by Howard and Padman (2021) could be used for this purpose. Introducing an artificial time lag and identifying the lag that maximizes correlation between the modeled tides and the GNSS data, following an approach similar to that described by Wild et al., 2025, would help to assess the magnitude and spatial variability of the viscoelastic response. Such an analysis could substantially strengthen the physical interpretation of the results presented in Fig. 9a.

**Minor comments:**

L14: Reword from "Changes at the boundary" to "Changes of the boundary" as the grounding line location generally serves as a proxy for stability considerations in numerical models.

L15: Terminology. Reword here and throughout from "ice-flexure zone" to "tidal-flexure zone"

L19: Terminology. Reword here and throughout from "Double Difference Interferometric SAR (DDInSAR)" to "Differential Interferometric SAR (DInSAR)". The word interferometric implies a (phase) difference between SAR images, the differential implies that two interferograms are being differenced. I know that DDInSAR can be found in the literature, but DInSAR is the modern terminology and the community strives to get rid of DDInSAR.

L23/24: Terminology. Reword here and throughout from "stable/unstable regime" to "steady/unsteady period". Stability of a glacier system is carefully defined in the ice-dynamics modeling community and we shouldn't use the term for other processes than MISI or MICI. Here, even if Milne Glacier experiences MISI, the analysis is concerned with the assumption of steady flow in DInSAR processing. In line with this, also reword "changes in glacier velocity were minimal" to "glacier flow was steady" and "more variable" to "unsteady". I also wonder if a

single observation of an unsteady period warrants the classification as a "regime" or "phase". I suggest rewording these terms to "period".

L32: Please replace the citation of the IPCC with original research in a discipline-specific journal like *The Cryosphere*.

L38: Change "migrating with tides" to "migrating with tides over short time scales (Freer et al., 2023)"

L41: Change "DDInSAR" here and throughout to "DInSAR". This means that "DSSH" can later be more intuitively compared to "DInSAR".

L43: "4 days" please compare to other often used SAR satellites like Sentinel-1, TerraSAR-X, COSMO, etc. Is this a dedicated mission, if so, please add this piece of information.

L43: Kim et al. 2024 use a terrestrial radar system, while the present study is about a space-borne system. Please clarify the difference or replace this reference with other DInSAR based studies of which there are many other relevant ones.

L46: Define "Hydrostatic line" here, as it is used later in the paper but I can't remember to have seen its definition.

L50: Terminology. Reword here and throughout "FZ span" to "tidal-flexure zone", and "GZ span" to "grounding zone" and refer to their width.

L55: Causes of complexity. The most relevant citations for the present study are Rack et al., 2017 and Wild et al., 2018.

L65: What "model datasets" ? It sets up the expectation of the reader to see some form of tidal-flexure modeling. Coming back to this after reading the paper, I think this is only GEM, right ? I think to warrant the "model datasets" here requires to perform the analysis at least with and without the GEM-derived IBE, in comparison with the GNSS analysis.

L67: Perfect location to add a statement about the quantification of the steady-flow assumption during times when it is clearly violated. I believe this should be the real focus of the manuscript.

L78: Add information about the tidal range and tidal regime. Is it a diurnal or semi-diurnal tide ? How long is the spring-neap-spring tidal cycle ? The introduction of the epishelf lake is of course interesting, but the space should be used for more relevant information to the paper, such as what is known about Milne Glacier's velocity variability, and how it motivates a hypothesis (I suggest to use the known velocity variability as a motivation to analyse how different viewing geometries are affected by horizontal flow variability).

L106: How many km upstream is the mentioned prograde bed slope ? How many years of average grounding-line retreat will it take to be reached ?

L108: Data and methods. Please add sub-section titles such as "Differential InSAR processing", "In-situ data", "Tide modeling", "Uncertainty analysis", "Grounding-zone width modeling", etc…

L111: A 10m spatial resolution is stated in the abstract, here 5m native resolution and the table shows about 2.5m raw pixel size. I know how these are affected by multi-looking and the topographic correction, so please describe these processing steps in more detail and state the chosen variables, which is also a great motivation to acquire a higher-resolution DEM to not lose any information in RCM.

L115: What seems to me the most important information is that orbit 9 has twice as large incidence angle as the other orbits. This should be better visible in the table and the description of the RCM data, which sets the paper up nicely for developing the unsteady-flow quantification or even the bending-effect hypotheses outlined in the major comments.

L121: reword to "effect of steady glacier flow".

L122: hydrostatic equilibrium line is not defined yet. Terminology. Also, please avoid confusion with the abbreviation of hinge line. Commonly HL denotes the hydrostatic line, and hinge line is spelled out.

L123: Terminology. Reword to "landward and seaward limits of the tidal-flexure pattern". The boundaries normally refer to already delineated GL and HL. Also, reword "glacier flexure" to "tidal flexure". I understand that the main contributor to "glacier flexure" here is the IBE, but the geophysical process is commonly called tidal flexure, and the IBE is one contributor (aside ocean tides and tidal loading on the Earth's crust, which is neglected here, right ?).

L126: To better distinguish between DInSAR images (the double-difference result) and times of single tide snapshots, please reword here and throughout to "three or four consecutive SAR data acquisitions" or "SAR scenes" to underline the snapshot character of the SAR data.

L129/130: First and only mention of phase-loss in lateral margins. Please add how ice-flow rotation leads to loss of coherence required for interferometry. Also, please read Wild et al., 2025 how weakening of ice properties in these zones affect stiffness and thus the flexural response of the entire glacier. This signal provides an exciting avenue for further research on bulk ice stiffness and buttressing. This section also lacks information of phase unwrapping and the coordinate used as a reference, which is also not shown in the figures.

L142: How were phase discontinuities corrected ? Only along the transects ? This and the mention of division by cosine of the incidence angle to get displacement should be moved to a dedicated DInSAR processing subsection in the methods.

L154: The correct references for the importance of atmospheric pressure are Rignot et al., 2000 and Padman et al., 2003.

L156: t-tide analysis. Please quantify how accurately the t-tide prediction replicates the input data, and how well it fits to an independent tidal model (such as Gr1kmTM). How do these uncertainties affect the results ? Is there a time delay between modeled tides and GNSS displacement ?

L157: Reword "corrected" to "added to" the IBE, since atmospheric pressure variations are a contributor to the tidal forcing (sum of ocean tides, load tides, and the IBE). Also, please add the conversion factor, was it 1cm drop for every positive anomaly of 1hPa ? Were pressure anomalies calculated over the entire time span, or over shorter chunks ?

L160: What's the temporal resolution of GEM ? Fig. 2c indicates it is lower resolution than the AWS data. Was the AWS pressure record low-pass filtered to a common cutoff frequency before any gaps were filled to account for any differences in temporal resolution ?

L162: Please quantify "good agreement" between GEM and AWS data. For a direct comparison, the AWS records will need to be low-pass filtered with the cutoff frequency of GEM before any correlations and RMSE can be calculated.

L163: Reword "DDInSAR result" to "DInSAR image" and also change to "SAR data acquisition" or "SAR scenes".

L164: Reword "DDInSAR subtraction", which implies a triple difference to "DInSAR combinations", which I think is not meant here.

L165: Where along each profile is DInSAR-derived surface height taken ? Normally a "freely-floating area" is averaged over, but this reads as if the far left side of each profile was taken ? To minimize the effect of interferogram noise, I suggest averaging over everything larger than 47 km distance, or even better any DInSAR displacement beyond the delineated hydrostatic lines.

L171:  DInSAR images contain information about the net sum of all tidal constituents (plus IBE and tidal loading) so fitting with a single constituent is physically invalid. Please refer to the methodology presented by Minchew et al., 2016 how to deal with apriori information on each sinusoid before the inversion, or employ an alpha map based approach for the unraveling to single tides.

L172: Please reword here and throughout from "SAR acquisitions" to "SAR data acquisitions" or "SAR scenes".

L174-176: "We used SSH…" I do not follow this sentence, please rephrase and say what hypothesis is investigated and how it is done.

L176: Reword "grounding-line delineations" with "tidal flexure zone delineations".

L180: Terminology: Reword "SAR observations" to "SAR data acquisitions" or "SAR scenes". The word observation normally refers to processed DInSAR images.

L183: Do the three hour GNSS acquisition period introduce a tidal bias in the derived mean daily velocity ?Is the error of 1.5 cm considered in the uncertainty analysis ?

L190: Quantify "agreed well" with an RMSE. Looking at Fig. 2a this is only true for the steady periods but the agreement isn't that good during summer and into the unsteady period.

L192: Reword from "effect of velocity" to "effect of velocity variability"

L201: Please label "Purple Valley" in Fig. 1

L204: Reword "DDInSAR observations" to "SAR data acquisition times for four different orbits".

L209: Please add the percentage of missing GNSS records of the total period of investigation.

L211-218: Very nice. It would have been beneficial to also conduct this part of the error analysis with the SAR-derived velocity fields and analyze any spatial patterns of the steady flow assumption.

L236: Do the delineations of grounding-zone width need to be corrected for the "bending effect" to be comparable to the theoretical width determined by Tsai and Gudmundsson ? In other words, is this a model for neutral layer displacement ?

L240: reword "RCM images" to "SAR-data acquisitions" or "SAR scenes".

L245: This is not surprising, given that ice stiffness controls the width of the tidal-flexure zone.

L254: Reword "during the last part of our intensive observation period" to " towards the end of the observation period".

L263: Reword "atmospheric pressure" to "IBE" or "atmospheric pressure variability", which is what is causing this signal.

L267-269: "These correlations…" move to Discussion.

L269-272: I think that one must add that the result of a double difference can have a larger absolute value than any of the individual values. So a 2.3 fold increase is essentially a measure of the non-linearity of the investigated process (a tiny change at the times of the SAR-data acquisitions will result in a much larger value in the DInSAR combination). So adding the IBE,

which is known to dominate in areas with small tidal range, will greatly improve any correlation with DInSAR measurements on the freely-floating part. Therefore, I think that it is useful to report on how the IBE improves the correlation at Milne Glacier, but it shouldn't be a main finding with a dedicated Figure in the main text. I suggest moving Fig.4 to the Appendix or SI, and cut back on the interpretation in the main text.

L272: Great that correlations improved after the IBE, but the RMSE is still huge when compared to the signal, about 15 cm of 50 cm DSSH. Please refer to the "tide-deflection ratio" approach outlined above on how to further minimize the RMSE to be within interferogram noise (<1cm).

L301: I don't quite agree with this interpretation (which also needs to be moved to the Discussion). The width of the flexure zone is independent of DSSH and only determined by ice stiffness. Also, the higher the DSSH, the more pronounced the spurious surface movement by the bending effect and thus the grounding-zone width.

L304-309: Nice quantification of the steady-flow assumption, which is the strength of the paper.

L314/315: Is this surprising ? Rather start this paragraph with why one would want to look at individual orbits and say that different incidence angles are more or less sensitive to horizontal flow variability.

L320: The different sensitivity of different orbits is interesting and additional evidence that the bending effect is at play. This signal should ideally be explored for a more glaciological research question.

L330: Is this a typo ? Orbit 43 here should be orbit 09, right ?

L331: Replace "different viewing geometry" to "higher sensitivity to horizontal displacement and variability thereof."

L333-346: I think this paragraph is speculative and the argumentation is invalid for the reasons outlined above and should therefore be removed. At a minimum, it should be moved from the results section to the Discussion section. The part talking about grounding-zone width (here distance between HL and point A) requires a dedicated investigation of the tidal forcing at the times of SAR data acquisition, which are here presented in their DInSAR combinations.

Similar to the width of the tidal-flexure zone (here Flexure zone span) and differential tidal displacement, where a linear regression over all the data seems physically unmotivated. The trends look statistically insignificant. I think this also presents either biases during the times of SAR-data acquisition or a temporal signal (which then can be linked to the emergence of new pinning points).

L351-367: This reads all like Discussion and should be moved out of the results section.

L355: Wait, I thought the observations are not aligned with the theory by Tsai and Gudmundsson ? What did I miss ?

L374: Here we switch from m/d to m/yr. Please choose one unit throughout the manuscript (text and figures) so velocities can be easier compared between satellite and GNSS.

L377: A 2.7 fold increase between flexure-zone and grounded glacier speeds. This is more than the temporal acceleration observed in the GNSS data (eye-balled from Fig. 2a about 2.1 fold). Taken together, these observations indicate temporal variability of longitudinal strain rates. Was this signal investigated with respect to the emergence of new pinning points and short-term variability in buttressing strength? This provides another exciting avenue for this manuscript as spatial patterns could also be investigated with the data at hand.

L386: Typo. "Fig. 11a" should be "Fig. 11c", right ? Also, the claim about larger ice discharge in the west requires calculation using the ice thicknesses shown in Fig. 10 and surface velocities.

L.387: Typo. "Fig. 11 b,c" should be "Fig. a,b", right ?

Discussion:
I found this section underwhelming as it's largely a summary of the presented results, without a clear evaluation of a glaciological hypothesis. I provided a few different avenues above and hope that depending on the target journal, this section will be reworked with either a remote-sensing focus or glaciological focus.

L400: Reword to "atmospheric pressure variability"

L410: Reword "pinning points drastically affected" to "pinning points control"

L421: I suggest starting this paragraph with how DInSAR assumes steady flow to isolate the differential vertical component commonly referred to as a fringe pattern delineating the tidal-flexure zone. Then state how the present study quantifies this assumption with available GNSS data. The difference for different orbits then depends on the sensitivity to horizontal flow variability, which should be investigated with the available SAR-derived velocity fields.

L429: Typo. "39.6 deg" is "39.5 deg" in Tab. 1

L431: Replace "viewing geometry" to "sensitivity to horizontal flow variability"

L435: Regarding the loss of coherence. Did you try to mask only ice-covered areas when matching slcs (the step before calculating interferograms). ? I found that this often greatly increases coherence compared to offset fitting the entire scene.

L437: Why are more in-situ velocity observations required ? I understood that the beginning of this paragraph that even though the velocity variability is large, the spurious signal in DInSAR is

small compared to the tidal forcing. I would have appreciated a statement how well only using SAR-derived velocity fields from speckle tracking can be used for estimating the spatial patterns of the steady flow assumption.

L443: I think this is the bending effect, it's just more pronounced if the background flow is elevated. Another motivation for using the 2D velocity fields for a detailed analysis.

L450: This sentence is quite speculative as the observations only show that the grounding line has retreated and acceleration is non-linear, but not that it's a runaway effect (which would require a numerical model of viscous ice-dynamics).

L494: I want to encourage the authors to perform the suggested further investigation of viewing geometry.

Data availability:
I understand that raw data are often subject to license agreements and can therefore not be published, but how about the processed InSAR and DInSAR images ? I want to encourage the authors to provide them (together with the datetimes of the corresponding SAR scenes) to the scientific community.

Where is the IceBridge data available. Please provide a link and DOI.

The modern standard is to provide field-data sets to the scientific community through an online provider (PANGEA, USAP-DC, NSIDC, etc) about 2 years after they've been acquired. And not by request to the corresponding author.

References:
Unify the formatting.
Schoof, 2007 seems missing.

**Figures**

Figure 1:
I suggest adding two panels with the mean velocity fields over the steady and unsteady periods. Or the steady one, with the second showing the spatial variability of percentage acceleration during the unsteady phase. Also, add the extent of panel b to panel a. And the location of GNSS and the phase unwrapping point to panel b. Two arrows with flight direction and azimuth of the different orbits might also be useful to estimate the sensitivity to horizontal flow variability.

Figure 2:
Panel (a): What is going on with the SAR-derived velocity spike in 2024-01-02 ? It seems to overlap with an increase in air temperature - is this a real signal ?

Panel (c): Please change the pressure unit from "kPa" to hPa", which is the standard practical unit

Figure 3:
Panel (d): Please add a unit to the colorbar or in the caption. I think these are radians. It looks like there is a nice freely-floating area towards the end of the profiles, so please average the DInSAR displacement over some distance before the comparison to DSSH.

Figure 4:
Move out of main text.

Figure 5:
Correcting the flexure curves for the "bending effect" would eliminate the spurious grounding-line migration within one local ice thickness (about plus/minus 300 m). Also, some flexure curves show distinct bumps upstream of the grounding line, which are indicative of viscoelasticity or grounding line migration between the SAR-data acquisitions. Why are the different shapes of the flexure curves not being discussed with respect to the timing (or tide, IBE) at the times of their underlying SAR scenes ?

Figure 6:
Again, the main take-away is that including the IBE improves the correlation but the large RMSE are kept. Does this Figure need to be shown in the main text ?

Panel (a): cyan label: Reword to "Reconstructed SSH (tide-based only)" and black label to "Reconstructed SSH (tides plus IBE)"

Panels (b-e): Are unraveled (or single-tide) or DInSAR combination results plotted against each other ? I think it should be single tides, so the x-values would correspond to the red curve in panel a ? Is the x-label incorrect ?

Please swap the order of panel b/c and d/e, to match the description in L173 and logic of Fig.4

Figure 7:
Is this separating parts of Fig. 4 d/e into steady/unsteady periods as defined by GNSS data as colors ? Another strength of the presented analysis, and I think the whole story about including the IBE to improve the statistics can easily be moved here by including two additional panels and moving Figs. 4 and 6 to the Appendix/SI.

Figure 8:
I don't think this figure adds to the storyline and can be summarized when presenting Fig. 7 in the main text. It should therefore also be moved to the Appendix/SI.

Figure 9:
For the reasons outlined above, temporarily integrating the data points by fitting a linear regression isn't physically motivated in the manuscript's current form. I think the panel (a) shows a signal that can be further explored, but since it doesn't add to the story Fig. 9 should therefore be removed from the manuscript.

We know from Schmeltz et al., 2002, and others that it's non-linear.

Figure 10:
I was a bit confused if flexure zones or grounding zones are displayed in panel a and b-d ? I think it's grounding zone limits, please clarify.

Figure 11:
Add "Hofmöller diagrams" at the start of the caption. Unify the labels to either m/d or m/yr. Also, add the location of the GNSS to one of the transects (consider a colored scatter with the same colormap, or its difference to the SAR-derived velocities through time). Interestingly, the acceleration in Jan/Feb 2024 occurs through the tidal-flexure zone. Was the signal investigated ?

**References**

Chien, Y., Zhou, C., Sun, S., Chen, Y., Wang, T., Zhang, B., 2026. Ephemeral grounding on the Pine Island Ice Shelf, West Antarctica, from 2014–2023. The Cryosphere 20, 245–263. https://doi.org/10.5194/tc-20-245-2026

Freer, B.I.D., Marsh, O.J., Hogg, A.E., Fricker, H.A., Padman, L., 2023. Modes of Antarctic tidal grounding line migration revealed by Ice, Cloud, and land Elevation Satellite-2 (ICESat-2) laser altimetry. The Cryosphere 17, 4079–4101. https://doi.org/10.5194/tc-17-4079-2023

Han, H., Lee, H., 2015. Tide-corrected flow velocity and mass balance of Campbell Glacier Tongue, East Antarctica, derived from interferometric SAR. Remote Sens. Environ. 160, 180–192. https://doi.org/10.1016/j.rse.2015.01.014

Li, T., Dawson, G.J., Chuter, S.J., Bamber, J.L., 2023. Grounding line retreat and tide-modulated ocean channels at Moscow University and Totten Glacier ice shelves, East Antarctica. The Cryosphere 17, 1003–1022. https://doi.org/10.5194/tc-17-1003-2023

Minchew, B.M., Simons, M., Riel, B., Milillo, P., 2017. Tidally induced variations in vertical and horizontal motion on Rutford Ice Stream, West Antarctica, inferred from remotely sensed observations. J. Geophys. Res. Earth Surf. 122, 167–190. https://doi.org/10.1002/2016JF003971

Padman, L., Erofeeva, S., Joughin, I., 2003. Tides of the Ross Sea and Ross Ice Shelf cavity. Antarct. Sci. 15, 31–40. https://doi.org/10.1017/S0954102003001032

Picton, H.J., Stokes, C.R., Jamieson, S.S.R., Floricioiu, D., Krieger, L., 2023. Extensive and anomalous grounding line retreat at Vanderford Glacier, Vincennes Bay, Wilkes Land, East Antarctica. The Cryosphere 17, 3593–3616. https://doi.org/10.5194/tc-17-3593-2023

Rack, W., King, M.A., Marsh, O.J., Wild, C.T., Floricioiu, D., 2017. Analysis of ice shelf flexure and its InSAR representation in the grounding zone of the southern McMurdo Ice Shelf. The Cryosphere 11, 2481–2490. https://doi.org/10.5194/tc-11-2481-2017

Ramanath, S., Krieger, L., Floricioiu, D., Diaconu, C.-A., Heidler, K., 2025. Automatic grounding line delineation of DInSAR interferograms using deep learning. The Cryosphere 19, 2431–2455. https://doi.org/10.5194/tc-19-2431-2025

Reeh, N., Christensen, E.L., Mayer, C., Olesen, O.B., 2003. Tidal bending of glaciers: a linear viscoelastic approach. Ann. Glaciol. 37, 83–89. https://doi.org/10.3189/172756403781815663

Rignot, E., Padman, L., MacAyeal, D.R., Schmeltz, M., 2000. Observation of ocean tides below the Filchner and Ronne Ice Shelves, Antarctica, using synthetic aperture radar interferometry: Comparison with tide model predictions. J. Geophys. Res. Oceans 105, 19615–19630. https://doi.org/10.1029/1999JC000011

Ross, N., Milillo, P., Nakshatrala, K.B., Ballarini, R., Stubblefield, A., Dini, L., 2024. Importance of ice elasticity in simulating tide-induced grounding line variations along prograde bed slopes. https://doi.org/10.5194/egusphere-2024-875

Schmeltz, M., Rignot, E., MacAyeal, D., 2002. Tidal flexure along ice-sheet margins: comparison of InSAR with an elastic-plate model. Ann. Glaciol. 34, 202–208. https://doi.org/10.3189/172756402781818049

Vaňková, I., Nicholls, K.W., Corr, H.F.J., Makinson, K., Brennan, P.V., 2020. Observations of Tidal Melt and Vertical Strain at the Filchner‐Ronne Ice Shelf, Antarctica. J. Geophys. Res. Earth Surf. 125, e2019JF005280. https://doi.org/10.1029/2019JF005280

Wild, C.T., Alley, K.E., Muto, A., Truffer, M., Scambos, T.A., Pettit, E.C., 2022. Weakening of the pinning point buttressing Thwaites Glacier, West Antarctica. The Cryosphere 16, 397–417. https://doi.org/10.5194/tc-16-397-2022

Wild, C.T., Drews, R., Neckel, N., Lee, J., Kim, S., Han, H., Lee, W.S., Helm, V., Rosier, S.H.R., Marsh, O.J., Rack, W., 2025. Monitoring shear-zone weakening in East Antarctic outlet glaciers through differential InSAR measurements. The Cryosphere 19, 4533–4554. https://doi.org/10.5194/tc-19-4533-2025

Wild, C.T., Marsh, O.J., Rack, W., 2019. Differential interferometric synthetic aperture radar for tide modelling in Antarctic ice-shelf grounding zones. The Cryosphere 13, 3171–3191. https://doi.org/10.5194/tc-13-3171-2019

Wild, C.T., Marsh, O.J., Rack, W., 2018. Unraveling InSAR Observed Antarctic Ice-Shelf Flexure Using 2-D Elastic and Viscoelastic Modeling. Front. Earth Sci. 6, 28. https://doi.org/10.3389/feart.2018.00028

Wild, C.T., Marsh, O.J., Rack, W., 2017. Viscosity and elasticity: a model intercomparison of ice-shelf bending in an Antarctic grounding zone. J. Glaciol. 63, 573–580. https://doi.org/10.1017/jog.2017.15